# Environment Actors Confronting a Post Climate-Related Disaster Scenario: A Feasibility Study of an Action-Based Intervention Aiming to Promote Climate Action

**DOI:** 10.3390/ijerph18115949

**Published:** 2021-06-01

**Authors:** Neide P. Areia, Alexandre O. Tavares, José Manuel Mendes

**Affiliations:** 1Centre for Social Studies, University of Coimbra, Colégio de S. Jerónimo, Apartado 3087, 3000-995 Coimbra, Portugal; atavares@ci.uc.pt (A.O.T.); jomendes@fe.uc.pt (J.M.M.); 2Department of Earth Sciences, Faculty of Sciences and Technology, University of Coimbra, 3030-790 Coimbra, Portugal; 3Faculty of Economics, University of Coimbra, 3004-512 Coimbra, Portugal

**Keywords:** climate change, action-based communication intervention, attitude change, feasibility study

## Abstract

(1) A climate change awareness-action gap still prevails in our society, with individuals showing significant social inertia regarding environmental issues. The communication on climate change is pointed out as one of the causes of the social inertia; (2) Following an action-oriented transformation research, the main goal of this study was to ascertain the feasibility of an action-based communication intervention, based on the participants’ exposure to a post climate-related disaster scenario. The feasibility of the action-based communication intervention was assessed in a focus group meeting, whose content was qualitatively analysed; (3) The results of this study demonstrate that action-based communication interventions are feasible to trigger an attitude transformation, catalysed by the experiential processing of climate change and by the affect arousal; (4) This study comes to contribute to a transformation of the climate change communication praxis, by highlighting the urgency to shift the for a new paradigm of communicating climate change, in which the affect and the experiential processing should account for conveying environment-related information in order to promote society’s climate action.

## 1. Introduction

2019 became the year of ‘climate emergency’ declarations. At a scientific level, more than 11,000 scientist signatories from around the world, banded together to warn humanity that “*clearly and unequivocally (…) planet Earth is facing a climate emergency*” [1]. Aligned with science, an exponential rise of ‘climate emergency’ declarations were observed at a political level, with more than a thousand local governments declaring this emergency in 25 countries [2]. However, only relying on governments and scientific institutions to combat climate change, may lead to a failure on these institutions’ effort to do so [3]. Scientists and policy makers have come to recognise that climate change urges climate action in several societal spheres, including from common citizens. Yet, it has seemed challenging to enhance the societal transition to environmental sustainability, as the general public is often considered the most difficult to transform [4]. Indeed, a substantial social inertia regarding the actual environmental crisis prevails in our societies, in which is obvious an insufficient public mobilisation and engagement to climate change [5].

At this point, a shift from a reductionist to a complex understanding of climate change is critical, with the Intergovernmental Panel on Climate Change (IPCC) having broadened its focus to consider the social factors—e.g., the potential mediator role of psychosocial variables in catalysing social climate action—related to climate change in its fifth report [6]. However, more than a half decade after the report’s publication, and even though society has been facing a climate emergency crisis [1], social-environmental research is still scant [7] and has been failing to precisely determine which (and how) variables may enhance individuals’ climate action and mitigate the actual social inertia regarding climate change [8,9].

The social inertia related to climate change may be explained by its distinctiveness as a risk, as it is difficult for individuals to experience it directly or detect it on a purely perceptual or sensory level [10]. The slow moving, cumulative, and unallocated nature of climate change, makes it an “abstract issue” for individuals [11,12]. For those reasons, society tends to regard climate change as a nonurgent and psychologically distant risk—spatially, temporally and socially—which, naturally, hinders the action mobilisation towards the problem [12,13]. On this matter, research continues to prove that individuals, even with knowledge on the subject, still rank climate change as a low priority issue compared to other societal risk issues that are easier to subjectively perceive (e.g., economy) [11]. It seems, therefore, that the simple knowledge on climate change accounts infinitesimally for the variance of public concern and thus for triggering climate action [14].

It seems obvious that an attitude change regarding the environmental issues goes beyond individuals’ cognitive dimension, by encompassing several key dimensions (e.g., psychosocial), usually neglected by the traditional sources of information (e.g., media) [10,15]. Taking this into account, this study aims to assess the feasibility of an action-based communication intervention, strategically designed to tackle the psychosocial dimensions of the human behaviour (e.g., affect), to further explore those dimensions’ role in facilitating individuals’ climate action.

## 2. Materials and Methods

### 2.1. Study Design

The study design follows the international guidelines (cf. 6SQuID) to develop quality community interventions, specifically the stage regarding the intervention’s feasibility testing on a small scale [16]. For this, two main sequential steps were carried out: (1) action-based fieldwork (action-based communication intervention), and (2) focus group to assess the intervention feasibility (Figure 1).

### 2.2. Participants

A total of 20 participants from five different European countries participated in this study: England (*n =* 2), Ireland (*n =* 3), Spain (*n =* 3), France (*n =* 7) and Portugal (*n =* 5). In order to assess the role of psychosocial variables (e.g., experiential processing) in enhancing an attitude change regarding climate change, an advanced or expert knowledge on the topic was assured in participants’ recruitment. For such, the inclusion criterion of the study was based on participants who were actively involved in environmental/climate change action or environment-related research. Regarding participants’ filiation, 9 (45%) were academics, conducting their environment-related research in universities, while the remaining 12 (65%) were stakeholders, taking their actions on environment-related technical institutions. A heterogeneous sample, regarding participants’ expertise was assured. The participants provided written informed consent prior to research activities. The subjects’ participation in this research was voluntary and participants were free to discontinue participation at any time. Table 1 displays the characteristics of the participants.

### 2.3. Action-Based Fieldwork (The Action-Based Communication Intervention)

The action-based fieldwork was carried out in regard to the action-based communication intervention. The intervention consisted of the participants’ exposure to a post climate-hazard scenario, specifically a *post* fire scenario in Pedrógão Grande. Pedrógão Grande is a Portuguese municipality in the centre of the country that was devastated by a forest fire between 17 and 24 June 2017 (Figure 2).

Considering the Forest Fire Research Centre report [17], 9672 ha. were burned, 65 people died, among them 8 children aged ten or under. More than 200 people were seriously injured. Of the 65 dead, 47 died on the EN236-1 road, the majority inside their cars (30). The EN236-1 is now known as the *Death Road* (Figure 3). Almost 200 people were additionally evicted from their houses.

The participants visited the scene of the fire on 17 October 2018, more than a year after the fire’s occurrence. The action-based communication intervention was facilitated (1) and mediated (2, 3, 4) by a trained sociologist for the purpose. The intervention consisted of:(1)*Visit to the municipality’s burnt area, to the Death Road and to the Grief Tribute Monuments built by the community to pay honour to the dead and victims of the fire*, in which participants could observe the physical impacts of the forest fire in vegetation, remains of the imprints of the burnt cars in the asphalt and grief-related tribute objects and monuments (e.g., flowers, toys) to the dead and a discussion with the sociologist that covered these topics.(2)*Visit to the Victims Association of Pedrógão Grande Forest Fire*, in which several Association members and their leader presented the emotional (e.g., grief, trauma) and pragmatic impacts (e.g., eviction cases) of the forest fire on the community.(3)*Meeting with a local policy maker*, in which the political perspective about the forest fire was presented. Participants could clarify some of the political management actions *pre-* and *post-*fire and discuss the public policies related to risk management.(4)*Meeting with an expert in GIS* (Geographic Information System tools) *development tools*, in which a GIS expert presented an online platform developed to optimise risk management responses regarding climate-related hazardous processes. Within this activity, participants discussed the relevance of GIS tools on climate change risk management.

### 2.4. Focus Group

The feasibility of the action-based communication intervention was verified, in one focus group meeting, that lasted approximately two hours. The focus group occurred the day after the action-based communication intervention.

The 20 participants were invited to take part in an experiential debriefing meeting—in focus group format—aimed at reaching a collective interpretation of the Pedrógão Grande community’s adaptation to the climate-related hazard. Specifically, the participants were encouraged to share their feelings about the day before experience and how the fieldwork possibly changed their perspectives on climate change, related hazardous and climate change risk management.

With the consent of all participants, the discussion was audio-recorded to be transcribed. The discussion transcription was uploaded for an in-depth analysis into NVivo^®^ qualitative data analysis software (QSR International PTY Ltd. Version 12, 2018, London, UK). Based on Braun and Clark’s (2006) recommendations [18], a thematic analysis of the data was conducted, following these procedures: (1) the transcript was read independently by the research team to enhance the immersion and familiarity with the collected data (NPA, AOT, JMM); (2) line-by-line coding initiated with open coding (NPA, AOT); (3) open codes were combined into overarching themes and subthemes (NPA, AOT); (4) preliminary themes and subthemes, initial description, and corresponding textual support were presented to the other members of the research theme (AOT, JMM) for final refinements.

The aforementioned process followed the standards of the COnsolidated criteria for REporting Qualitative research (COREQ) guidelines [19].

## 3. Results

The analysis of the focus group yielded 5 main themes, some comprising several subthemes. The founded themes represent key stages of an attitude change process regarding climate change. These key processes are presented below and additional example quotes supporting each theme/subtheme are shown in Table 2.

### 3.1. Acknowledgment

This theme corresponds to the participants’ recognition and cognitive learning from the exposure to post-climate hazard scenario, specifically regarding its predictors (e.g., failures of risk management) and effects (e.g., impacts on the community).

#### 3.1.1. Risk Management Failures

Participants discussed the *risk management failures* that could have influenced not only the management of the wildfire process, but also the community’s adaptation. The lack of articulation between all interested actors (e.g., stakeholders, civil society, policy makers) in managing the wildfire and the *post*-fire adaptation was repeatedly emphasized by the participants (example quote on Table 2).

Still, some participants mentioned the lack of communities’ education to properly act when facing a forest fire, such as not striving to escape from the fire, instead searching for a body of water (e.g., water tank) or the need to evacuate their houses and how to properly do so.


*“How is possible that people are not alerted and taught how they can escape from the fire? (…) I’ve realised that people want to stay close to their houses. But we must have tools to predict this kind of situation—like the platform that the other guy was showing—and thus inform the communities in order to make people act without putting their lives at risk.”*
(p. 7)

Also mentioned was the lack of resources to prevent climate hazards (e.g., promote land clearing, promote the plantation of less flammable vegetation, improve the risk communication plan) and to provide further support to the affected communities (e.g., house rebuilding, provision of emotional support for the bereaved).


*“… it seems that there’s no income to help people from there. Indeed, there could be better designed models to predict and adapt to these kind of situations, but we are always facing and fighting against the same problem: the lack of money.”*
(p. 18)

#### 3.1.2. Impacts

With regard to the impacts, the participants frequently highlighted the emotional impacts on the survivors, especially on the bereaved, as it can be read on the example quote on Table 2.

Some mentioned the physical impacts of the scenario, especially regarding the remaining burnt trees and the actual deforestation that the wildfire caused (“*That scenario of total devastation made me realise that climate change is definitely a real problem that we have to actually face.”* p. 3)

Regarding the victims, the dead children were particularly referred (*“It was so hard to go there* [referring to the ‘Death Road’] *and realise that little children died burnt there.”* p. 4).

#### 3.1.3. Community Adaptation

The community adaptation after the wildfire, was particularly discussed throughout the focus group meeting, since the community was steadily mentioned as an example of proactivity and resilience. For instance, the quick establishment of a social movement to help in the deceased’ identification and to further provide instrumental and emotional support to the bereaved and other survivors, was deeply felt by the overall group.


*“A few things that touched me yesterday: the importance of the local level and particularly the role of that community network after the disaster”*
(p. 5)


*“For me, the most impressive thing from yesterday, was the courage of the people of that community, the courage of [Victims Association member name], in promptly organizing themselves.”*
(p. 11)

### 3.2. Affect Arousal

This key process relates to the overall affect experienced by participants after the exposure to the climate hazard scenario. It entails an immediate affective response and a more complex phenomenon of internalization and projection to the disaster’s context (e.g., family context, professional context).

#### 3.2.1. Immediate Affective Response

Most participants described negative emotions, such as shock, surprise or even grief, when exposed to the disaster scenario or visiting the Victims’ Association, as shown in quote example in Table 2.

However, it is worth mentioning that two participants had a positive affective response regarding the exposure, finding it meaningful and motivating for doing further work and research on environmental issues.


*“Sometimes, we may feel that what we are doing or researching is not valuable or work at all. When we are working, we must find out the meaning of what we are doing, our direction. And what I saw yesterday, helped me to find a meaning to what we are doing.”*
(p. 15)


*“I think that beyond the shock regarding the situation, we have seen the light in the tunnel. We now know the gaps in our work, and we can think about the things we may change in the future.”*
(p. 16)

#### 3.2.2. Internalization & Projection

Remarkably, more than half the participants projected the experience of the community and survivors to their own lives, especially—and more frequently—for their family and social networks contexts (*“I couldn’t avoid thinking about my family and my children, as everyone here I guess. This is the ultimate risk of climate change: losing our families”* p. 13). A few other participants projected the experience to their own professional context (*“Here, with this local community, we can see the crossing, the communalities, between the farmers in France with whom I work everyday. And with me, too. With all of us, I guess.”* p. 10).

### 3.3. Discourse Deconstruction

The coding of this theme considered the discourse typology of the risk categorization model, developed by Renn (2008) [20]. This theme represents the risk discourse evolution identified in participants’ speech, specifically from an instrumental/epistemological to a participative/reflective discourse.

#### 3.3.1. Instrumental and/or Epistemological

Only two participants showed an *instrumental and/or epistemological discourse*, in which a dependency could be identified on the political system to manage risk events or in their own field of research, without considering the other interested actors’ role in the assessment and management of risk (see quotes examples in Table 2).

#### 3.3.2. Reflective and/or Participative

In contrast, the majority of participants seem to have recognised the role of the community, common citizens and other scientific areas (e.g., social sciences) in the assessment and management of risk and thus have shown a more *reflective and/or participative discourse*.


*“What I learnt yesterday: the resilience of communities. Actually, they are the ones that respond first. So, the discussion we have around the catchments and how we manage the catchments, will mean nothing if we don’t have the community in that discussion.”*
(p. 2)


*“We need to get our communities involved (…) We can’t wait for the State decide for us. Instead, we need to get more involvement from the communities and municipalities to transform our practices.”*
(p. 11)

Interestingly, a participant that had begun the discussion with an instrumental discourse, shifted his speech for a more participative discourse throughout the dialogue, in which he acknowledged the role of social sciences on climate change mitigation/adaptation.


*“I believe that sociologists are important in climate change projects. But I never thought about the psychologists. And today, with our discussion, I realised that they are important too. But there is no funding for that, I guess. How can we, with the communication we make, raise awareness for their importance?”*
(p. 9)

### 3.4. Action—Challenges

This theme represents a dialectical relation between the mobilisation to act/promote action and the acknowledgement of the challenges and difficulties to do so. Also, some solutions for the highlighted challenges, were co-reflected by participants.

#### 3.4.1. Mobilisation to Act or Promote Action

As a general rule, participants demonstrated an urge to act or change their previous behaviours (“After this, I’m sure that my own practices must be transformed.” p. 11) and to raise awareness and thus behaviour change for the others.


*“This gave me a lot of ideas and when I go back to Ireland I’ll talk to the farmers, with whom I work, about the future. Something must be done. I think farmers don’t realise that climate change is a real problem.”*
(p. 3)


*“My first reaction on going back to France will be to tell people: ‘don’t plant eucalyptus, it’s dangerous. It is making money in a short term, by causing a long term problem, such as a cancer’.”*
(p. 9)

#### 3.4.2. Acknowledgment of the Challenges to Promote Action

When the main difficulties to enhance the engagement of the civil society to climate change or to improve risk management models were discussed, several were pointed out by participants. For instance, the lack of resources (*“They do what they can, with the little resources that they have [regarding the social support team for the bereaved from Pedrógão Grande]; but the survivors would need a specialized trauma intervention.”* p. 19) and the communities’ culture were occasionally mentioned (*“Yesterday, with our reflection on society and how people react to this kind of situation, I could conclude that the old-fashioned values of communities are positively affecting the communities”* p. 1).

On the other hand, communication with communities was widely discussed throughout the meeting. Communication was frequently recognised as inefficient in engaging the civil society to climate change.


*“Were you aware of last years’ lack of water in Cape Town, South Africa? This was not very well reported around the world. In Portugal, last year’s fire, as well. (…) I don’t have much to say; the climate change communication is a shame. My reflection is: why do we have to get to this point before people start to listen? And it seems that we have to make people listen.)”*
(p. 1)

The communication between policy makers, stakeholders and communities was criticized by some participants (*“So we need to bring more bridges and connections between the communities and the other involved actors.”* p. 18). Likewise, science-communication was considered an area for improvement (*“As researchers in this field, I think we should be more accessible and facilitate that communication with others.”* p. 15).

With these communication gaps pointed out, some participants suggested that the role of technical institutions and/or stakeholders would be of utmost importance in conveying the science-based information to society (*“The University must be an open space and not a closed space. There are a lot of stakeholders that can mediate the dialogue between academics and communities.”* p. 16). Others, made reference to the usefulness of social media to engage citizens with climate change (*“If we want to have an impact on society, we may think about using the social media.”* p. 5; *“We must do something in France to improve my communication on these matters and raise awareness of people. I think Twitter and Facebook are important tools for that.”* p. 12).

### 3.5. New Meta-Narratives on Climate Change

New meta-narratives on climate change accounted for the acknowledgment that it is crucial to move forward from a global to a local perspective and the need to shift from the traditional approach to a more innovative approach for climate change.

#### 3.5.1. Local/Community Context Intervention

After the exposure experience, most of the participants acknowledged that the climate change and related-risk management approach should be conducted at a local level, in order to facilitate the active involvement of citizens.


*“Intervention must be at a local level, by developing local responses. These kind s of responses must be actions that people understand, recognise. So, we’d get people more involved on our adaptation and mitigation plans.”*
(p. 18)

#### 3.5.2. Approach Reframing

When discussing the most suitable comprehensive and intervention approach for climate change, participants agreed that climate change is a complex phenomenon with multidimensional predictors and impacts and, for such, it urges a shift on its traditional approach in order to efficiently respond to the problem.


*“About yesterday and about what we’ve been discussing here, I must say that we talked about soil, deforestation, water, catchments, and—totally unexpected for me—about the emotional things related to that fire. So, for me it’s obvious: climate change needs the cooperation of several scientific areas.”*
(p. 1)

Additionally, the inclusion of social aspects on the climate change approach (e.g., climate-hazards trauma related, communities’ culture) was emphasized by some participants that argued for the usefulness of these scientific areas for better coping with the problem.


*“What I realised and what I can conclude with the experience of yesterday is that climate change needs an interdisciplinary approach. When I started doing research in this field, I was always thinking ‘what is a psychologist doing on this research team? What is my role? How can I contribute?’ Yesterday, I realised that it is truly important.”*
(p. 19)

## 4. Discussion

An effective communication on climate change is considered pivotal to achieve an attitudinal shift of society towards a sustainable paradigm of thinking and acting [21]. However, communication on environmental issues seem to be failing to actively engage individuals to act against climate change [4,7,22,23]. For this, the aim of this study was to contribute to a transformation on climate change communication praxis, by suggesting the relevance of action-based climate change communication tools, aiming to facilitate the experiential processing of climate change, as feasible to increase risk perception and thus catalyse individuals’ attitude changes.

The findings from this study—specifically from the thematic analysis of the communicative focus group—yielded results that have a great deal of relevance for the climate change communication approach.

Particularly, action-based communication tools, grounded on the experiential processing of climate change, seem to be useful for enhancing individuals’ attitude change towards climate change, through an attitude transformation process, that comprised—in the case of this particular study—three key elements: *acknowledgment*, *affect arousal* and *risk perception*, which may be recognised in the participants’ discourse deconstruction, a dialectical relation between action-challenges, and the development of new meta-narratives on climate change. These processes seem to have been facilitated by an *experiential processing of climate change* (Figure 4). The attitude transformation process, enhanced by action-based communication tools, is further discussed.

As mentioned above, it was possible to detect indicators of an attitude shift in both environment-linked researchers and stakeholders after the exposure to a post fire scenario and the active interaction with the community affected by the fire. It is possible to relate the participants’ evolution on their risk discourse and their attitude change toward climate change, due to the experiential process of a climate-hazard situation. According to Beard and Thompson (2012) [24], the direct observation of climate change impacts is an important factor in raising individuals’ awareness on climate change. For that reason, we argue that the experiential process—facilitated by the action-based communication intervention—reduced the psychological distance of climate change and thus enabled the subjective perception of the problem as a real and imminent risk (e.g., p. 3), not only in geographically distant places, but also in their own contexts (e.g., p. 10), posing a real risk to their own and their relatives (e.g., p. 13). Beyond the acknowledgement of the reality of climate change and its impacts “here and now”, participants were able to recognise some of the risk management gaps and thus called for a risk management reform, such as, opening the dialogue to other fields of expertise (e.g., p. 19); considering the communities and citizens in risk management plans (e.g., p. 7); improving the articulation between institutions and the community and citizens (e.g., p. 16); and, specifically regarding the fire process, the introduction of different vegetation species to decrease the probability of fire (e.g., p. 9) or to properly educate citizens’ how to act in the face of a fire (e.g., p. 7).

Research on the effectiveness of action-based education methods, especially developed for school-aged people, has been proving the usefulness of these communication/education methodologies in actively engaging children and teenagers to climate change. For instance, Pruneau et al. (2003) [25], exposed a group of teenagers to the effects of climate change in two coastal communities. The authors verified that the experiment had improved the students’ conceptions of climate change, arguing that the local study of the phenomenon was crucial in raising awareness regarding climate change and its’ effects on coastal areas [25]. Other similar studies, based on action-based activities, also have proven efficacy in raising awareness and mobilizing individuals to act towards climate change [26,27,28]. This evidence comes to corroborate that a climate change communication framed to reduce psychological distance is an efficient strategy for increasing individuals’ engagement to environmental issues, as suggested by Jones, Hine, and Marks (2016) [29].

Distinctive from the aforementioned studies, we argue that the intense affect triggered by our action-based communication intervention, facilitated by its experiential processing dimension, also played a key role in the participants’ appraisal of climate change risks, in the increasing of their risk perception and, therefore, contributed as a catalysis of their attitude transformation towards climate change. The affect is classically defined as an associative, and automatic reaction that is subsequent to a cognitive representation of a stimulus and guides both the information processing and judgement and decision making [30,31]. Actually, in socio-environmental research, emotional arousal is steadily claimed as a key element to prompt citizens’ action towards climate change, being considered a powerful motivational force [12,32].

In our action-based communication intervention, the emotional arousal was clearly achieved through the exposure to the burnt area, the visit to “Death Road” and grief tribute monuments to the forest fire victims and through the interaction with the members of the Victims Association. The day after, in the communicative focus group, the majority of the participants reported an intense negative affect, such as shock (e.g., p. 9) and sorrow (e.g., p. 16), especially with regard to the burnt area and the survivors’ grief. These led to an increased concern for their own personal context (e.g., the risk of losing their loved ones due to a climate-hazard, e.g., p. 2) and, therefore, triggered mobilisation to act with regard to the own behaviours (e.g., p. 11) or to promote action in the others (e.g., p. 10), despite the acknowledged difficulties regarding the climate change communication (e.g., p. 16). These results are consistent with previous research, in which the experience of a negative affect regarding climate change impacts, was demonstrated to be related to an increased climate change risk perception and thus active engagement with the problem [10,13,33,34]. As such, based on prior evidence [10,13,33,34] and the particular role of the affect of arousal on this study, we argue that models of climate change communication should take into account this key element in order to increase climate change risk perception and an attitude transformation on the affected individuals. However, it is worth mentioning that the risk perception of climate change, is not automatically triggered by an effective response [35], as other variables play an important role on the individuals’ attitude change, as mentioned above.

Indeed, from this study, and as argued by Carbaugh and Cerulli (2012) [36] and Schweizer et al. (2013) [37], we may infer that the traditional climate change communication approach would benefit from a change of paradigm. New forms of communicating climate change should consider specific key elements, such as the affect arousal and the experiential processing of climate change. For achieving this, action-based communication tools for communicating climate change, should be developed and implemented not only in schools, but also in other civil contexts—such as the one in which we tested an action-based communication tool—as they prove to be feasible to catalyse an attitude transformation on individuals towards climate change.

### Limitations, Strengths and Future Directions

The main limitation of this study is related to the fact that it was not possible to ascertain how much the explored variables, such as the affect arousal or the subjective perception of climate change, explained the variance of climate change risk perception. As such, future studies should account for a mix-method design in order to quantitatively determine the effectiveness of action-based communication tools and the influence of specific factors (e.g., affect, socio-demographic characteristics) on climate change risk perception.

Additionally, this study was limited to a sample of European environmental actors, with relevant knowledge on climate change. Although previous studies have proven that knowledge about climate change does not have a direct effect in promoting behaviour change towards more sustainable patterns of living [10], results from this study should be interpreted with cautious taking into account a possible knowledge-related bias effect. Therefore, future studies should consider more heterogeneous samples, regarding other populations (e.g., school-aged people, vulnerable populations) and other cultures (e.g., non-Western cultures), in order to overcome the likely bias resulting from extensive knowledge about climate change.

If, on one hand, the selection of a sample consisting of environmental actors may be a limitation, on the other hand, it may also be considered a strength of this study. By guaranteeing the a priori participants’ knowledge on climate change, the role of other key variables (e.g., affect arousal, experiential processing of climate change) on enhancing an attitude change was better projected.

Finally, this study comes to question the traditional climate change communication praxis, by suggesting innovative ways to conveying climate-related information to the civil society that seem more effective in catalysing an attitude change that the old-fashioned ways.

## 5. Conclusions

Action-based climate change communication has the potential not only to improve the knowledge on climate change and raise awareness, but also to increase climate change risk perception and thus an active engagement with the problem. The affect arousal and the experiential processing of climate change seem pivotal in catalysing an attitude change and are largely facilitated by an action-based climate change communication. It is therefore suggested that the traditional communication praxis should be transformed, by introducing some of the suggested communication practices, in order to actively engage the civil society with one of the most challenging problems of this century: climate change.

## Figures and Tables

**Figure 1 ijerph-18-05949-f001:**
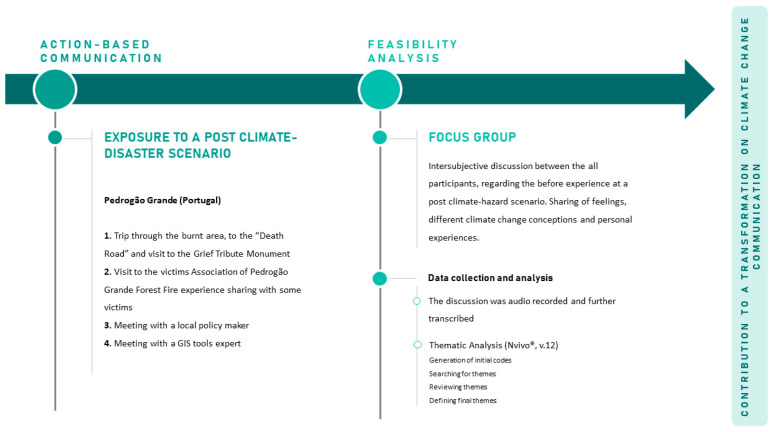
Study design.

**Figure 2 ijerph-18-05949-f002:**
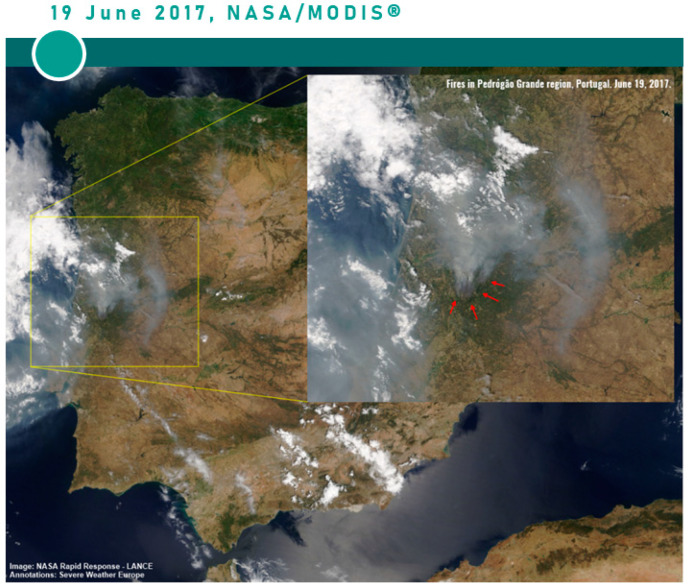
NASA/MODIS^®^ satellite image of the fires Pedrógão Grande in 19 June 2017.

**Figure 3 ijerph-18-05949-f003:**
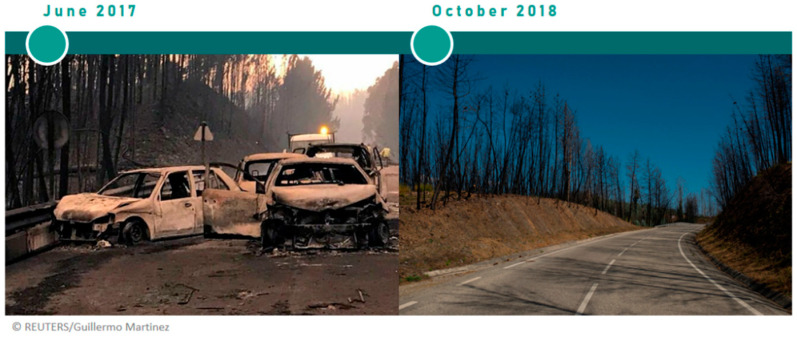
The EN236-1, *Death Road*, immediately after the forest fire in June 2017 (REUTERS^®^) and more than a year later in October 2018 (The provided pictures do not depict the exact kilometer of the EN236-1).

**Figure 4 ijerph-18-05949-f004:**
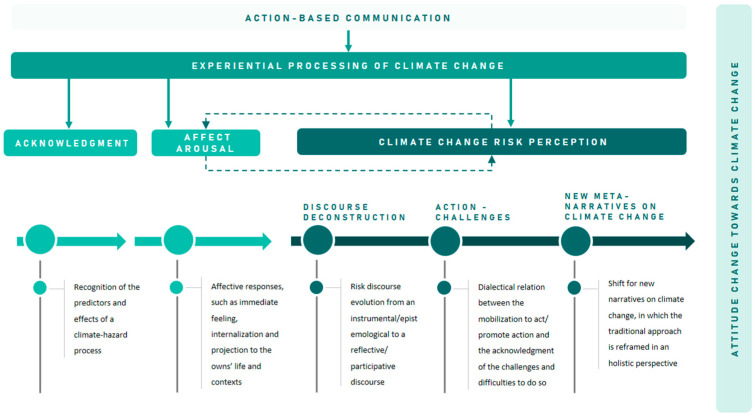
Climate change attitude change process enhanced by climate change action-based communication tools.

**Table 1 ijerph-18-05949-t001:** Characteristics of participants.

Code	Country	Field of Expertise
p. 1	England	River and Fisheries
p. 2	England	Biology River and Fisheries
p. 3	Ireland	Climate change, agriculture and food security
p. 4	Ireland	Economics Environmental economics
p. 5	Ireland	Economics Environmental economics
p. 6	Spain	Agricultural Engineering Agriculture ecosystems and crop production
p. 7	Spain	Environmental Sciences Remote Sensing and GIS specialist
p. 8	Spain	Agricultural Engineering GIS specialist
p. 9	France	Climatology
p. 10	France	Climatology GIS specialist
p. 11	France	Climatology
p. 12	France	Agricultural Engineering
p. 13	France	Agricultural Engineering
p. 14	France	Agricultural Engineering
p. 15	France	Agricultural Engineering
p. 16	Portugal	Geography Territorial policy
p. 17	Portugal	Geology Environmental geology
p. 18	Portugal	Mining engineering Risk management specialist
p. 19	Portugal	Psychology
p. 20	Portugal	Sociology

**Table 2 ijerph-18-05949-t002:** Attitude change process after a post climate-hazard scenario.

Main Theme	
Subtheme	Example Quotes
**1. Acknowledgment**	
Risk management failures (*n =* 13)	*“(…) So it was very confusing for me to have different ways of responding to these kinds of local issues (…) there’s a way framed in an institutional point of view and there’s and opposite one, of the individuals.”* (p. 18)
Impacts (*n =* 9)	*“(…) it’s actually what we could see: the grief of being the one who was left behind (…). She lost* [mentioned the death victims kinship with the survivor],* the whole family, indeed. And she could only feel guilty to have survived. She was grieving. That’s what I felt.”* (p. 2)
Community adaptation (*n =* 10)	*“They* [referring to the locals] *decided to start a social movement. They designed it, developed it and established it in the field. A social movement with the necessary links, with the supporters … that was clearly designed for that territory, for that context.”* (p. 18)
**2. Affect Arousal**	
Immediate affective response (*n =* 16)	*“Yesterday, it was very difficult for me to face that hellish scenario (…) I was touched by the situation, of course!”* (p. 12) *“I’ve never heard about a situation like this one; of what happened here. When I think about fire, I know that it goes fast. But I never thought that it would burn so many hectares of forest as this did. It was shocking for me and especially sad to talk with the survivor victims.”* (p. 9)
Internalization & projection (*n =* 10)	*“We always talk about climate change and economic losses—at least, it is how it happens in my field. We are always worried about losing the maize, the crop. We also talk about the environment risk, the river that has been polluted or the risk of flooding, of course! But I’d say that we have never talked about the ultimate risk of losing your family, your community. Since yesterday I see that the economic impact of losing a crop, is incomparable to the value of losing our family.”* (p. 2)
**3. Discourse deconstruction**	
Instrumental or Epistemological (*n =* 2)	*“(…) Indeed, the power is so strong in my country that it is impossible to implement adaptation actions by ourselves. It’s up to the State to so establish them.”* (p. 9) *“In my work, I design the climate situation for the future, in Andalucía. Further, it will be possible to develop tools to implement adaptation measures regarding climate change.”* (p. 6)
Reflective or Participative (*n =* 10)	*“What I learnt yesterday, was the resilience of communities. Actually, they are the ones that respond first. So, the discussion we have around the catchments and how we manage the catchments, will mean nothing if we don’t have the community in that discussion with community.”* (p. 2) *“We are frequently investigating to respond to our interests. But we should not forget our responsibility towards the society. We are doing this for them. (….) now we must make an effort to be more effective, to find better solutions (…) and be able to predict those areas at higher risk, to alert the population (…). I’m always thinking about erosion, tools, sensors, (…) and I realised yesterday that science is also humans, society, and the social aspects. This is also scientific work. Sometimes I don’t understand, I’m not able to understand. (…) but we need to understand how we can involve communities in our projects (…)”* (p. 7)
**4. Action—Challenges**	
Mobilisation to act or promote action (*n =* 16)	*“I think it must be us to first take some action and then, with our behaviours, influence other people. We also must promote local action, the protection of our rivers, actions to prevent the risk of fire or other hazards. (…) But we have to act now!”* (p. 10) *“We have to change the children; because they are the ones that will face this fatality.”* (p. 13)
Acknowledgment of the challenges to promote action (*n =* 28)	*“Communication is important, but a difficult matter. We must deal every day with our community (…) we have to replicate these best practices in risk communication and mitigation. In my institution we are concerned with our communication plan with the community, particularly with the farmers. We are always aiming to improve our relation. The [Institution name] is a present partner, where we can share our knowledge and, of course, learn more about the reality from those whom take care of the land. So we always have this on mind: we must break the academic walls and work with the community.”* (p. 16)
**5. New meta narratives on climate change**	
Local/community context intervention (*n =* 14)	*“So, what have I learnt from yesterday? We have the context that changes our intervention scope. In this case, it was the fire and that community’s specificities. The context changes how the recovery is, the responses, the involvement of the actors, the political action and so on. The strategy for one context does not serve for another context. So, the strategies must be designed at a local level, at a community level.”* (p. 18)
Approach reframing (*n =* 9)	*“Climate change is definitely a global and multidimensional problem and it needs to be considered interdisciplinary. Also, social perspectives, psychological perspectives must account for the climate change comprehension.”* (p. 19)

## Data Availability

The data presented in this study are available on request from the corresponding author. The data are not publicly available due to restrictions regarding research ethics.

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
