# Peer review of "Environment Actors Confronting a Post Climate-Related Disaster Scenario: A Feasibility Study of an Action-Based Intervention Aiming to Promote Climate Action"

_ijerph, 2021, doi:10.3390/ijerph18115949_

Round 1

Round 1

Reviewer Report

Dear authors, 

The article is of interest but requires a few changes to be made that are described below.

Abstract

The abstract has an Introduction, Design, Results, and Discussion. However, the mean aim of the research is not clear. Moreover, Value or Originality there are not as well. You should remember to provide readers an analysis of the value of your results. It’s a good idea to ask colleagues whether your analysis is balanced and fair. You can also conjecture the future steps of your research.

Materials and Methods

In the section of participants, Table 1 has as a heading the affiliation of the participants (Re-searcher or Stake-holder), however, it is not indicated. 

You should indicate how the participants' contribution was (compensated or voluntary).

In the section focus group, more details about the sections such as the number of sessions and the time these lasted.

Annex

A summary of the developed focus group guide containing a general listing of the topics that were addressed or questions that were asked, as well as the incorporation of new topics generated by the invitees. This would help other researchers replicate your research.

In general terms, I consider the article has a lot of potential. However, the aspects mentioned before are necessary to improve your paper. Please follow the recommendations.

Author Response

#Commentary 1: The abstract has an Introduction, Design, Results, and Discussion. However, the mean aim of the research is not clear. Moreover, Value or Originality there are not as well. You should remember to provide readers an analysis of the value of your results. It’s a good idea to ask colleagues whether your analysis is balanced and fair. You can also conjecture the future steps of your research.
#Authors’ response to commentary 1: Thank you for the valuable suggestion of improving the abstract in order to clarify the study main objectives and highlighting its potential. We have performed changes on the abstract in order to follow the reviewer’s suggestions. In the manuscript modified version, the abstract reads as follows:
“(1) Background: a climate change awareness-action gap still prevails in our society, with individuals showing a significant social inertia regarding environmental issues. The communication on climate change is pointed out as one of the causes of the social inertia; (2) Methods: following an action-oriented transformation research, the main goal of this study was to ascertain the feasibility of an action-based communication intervention, based on the participants’ exposure to a post climate-related disaster scenario. The feasibility of the action-based communication intervention was assessed in a focus group meeting, whose content was qualitatively analysed; (3) Results: results of this study demonstrate that action-based communication interventions are feasible to trigger an attitude transformation, catalysed by the experiential processing of climate change and by the affect arousal; (4) Conclusions: this study comes to contribute to a transformation of the climate change communication praxis, by highlighting the urgency to shift the for a new paradigm of communicating climate change, in which the affect and the experiential processing should account for conveying environment-related information in order to promote society’s climate action.”

#Commentary 2: In the section of participants, Table 1 has as a heading the affiliation of the participants (Researcher or Stakeholder), however, it is not indicated. You should indicate how the participants' contribution was (compensated or voluntary). In the section focus group, more details about the sections such as the number of sessions and the time these lasted.
#Authors’ response to commentary 2: Thank you for the of utmost importance suggestions to improve the methods sections. Modifications were performed in order to meet the reviewer’s expectations. 
• With regard to the participants’ filiation, this is clarified in the text, and reads as follows: 
“Regarding participants’ filiation, 9 (45%) were academics, conducting their environment-related research in universities, while the remaining 12 (65%) were stakeholders, taking their actions on environment-related technical institutions.”
This was also a problem reported by another reviewer. We agree that the information on the table may be redundant, as in text we provide the prevalence data on the participants’ filiation. Therefore, and as suggested, we have removed these data from the table, by removing the columns “Researcher” and “Stakeholder”.

• Concerning participants’ contribution to the research, this is now clarified in the new manuscript’s version (voluntary). Furthermore, we provided more information on the ethical procedures, particularly information on the informed consent: 
“Participants provided written informed consent prior to research activities. The subjects’ participation in this research was voluntary and participants were free to discontinue participation at any time.”
• Regarding the focus group session, we added more detailed information on the number of sessions (1) and its duration (approximately 2 hours). In the manuscript it reads, as follows: 
“The feasibility of the action-based communication intervention was verified, in one focus group meeting, that lasted approximately two hours.”

Reviewer 2 Report

The paper is well written. I suggest a few small changes:

line 35: "and policy makers have come to recognize"

line 37: "as common individuals the general public is"

Table 1: There is no information in the "Researcher" and "Stakeholder" columns. Is it missing? Above you state that 45% were researchers, and 65% were stakeholders. If you do not wish to show this data, then remove the columns.

Figure 3. Unfortunately the before and after photos were not taken in the same place. Not a serious problem, but perhaps that should be made clear in the caption.

Author Response

Author's Notes to Reviewer 2

#Commentary 1: line 35: "and policy makers have come to recognize".

#Authors’ response to commentary 1: Thank you for the correction. We have performed the suggested amendment in the revised manuscripts’ version.

#Commentary 2: line 37: "as common individuals the general public is"

#Authors’ response to commentary 2: Thank you for the correction. We have performed the suggested amendment in the revised manuscripts’ version.

#Commentary 3: Table 1: There is no information in the "Researcher" and "Stakeholder" columns. Is it missing? Above you state that 45% were researchers, and 65% were stakeholders. If you do not wish to show this data, then remove the columns.

#Authors’ response to commentary 3: Thank for the commentary/suggestion. This was also a problem reported by another reviewer. We agree that the information on the table may be redundant, as in text we provide the prevalence data on the participants’ filiation. Therefore, and as suggested, we have removed these data from the table, by removing the columns “Researcher” and “Stakeholder”.

#Commentary 4: Unfortunately the before and after photos were not taken in the same place. Not a serious problem, but perhaps that should be made clear in the caption.

Authors’ response to commentary 4: We added a footnote, in which we attempt to made clear that the pictures do not depict the exact same location in the EN236-1. In text (footnote), it reads as follows:

“The provided pictures do not depict the exact kilometer of the EN236-1”

Reviewer 3 Report

The authors have provided an interesting qualitative research about action-based intervention aiming to promote climate action. 
I think the authors could take a look at other published articles about formats and styles. For example, you do not need to write the abstract in point format. The template also provides instructions for reference/citation styles. 
The paper itself can also be improved in many ways. For example, the text in the figures is barely legible. Tables can also be properly formatted - e.g., in table 1, the last two columns are empty. 
There are also several questions related to the research design. For example, how to determine the sample representativeness? Are the participants selected randomly from different groups (e.g., researchers from certain fields or stakeholders). 

Author Response

#Commentary 1: The authors have provided an interesting qualitative research about action-based intervention aiming to promote climate action. I think the authors could take a look at other published articles about formats and styles. For example, you do not need to write the abstract in point format. The template also provides instructions for reference/citation styles. The paper itself can also be improved in many ways. For example, the text in the figures is barely legible. Tables can also be properly formatted - e.g., in table 1, the last two columns are empty. There are also several questions related to the research design. For example, how to determine the sample representativeness? Are the participants selected randomly from different groups (e.g., researchers from certain fields or stakeholders).

#Authors’ response to commentary 1: Thank you for the commentaries regarding the manuscript, which we attempted to respond.

  • With regard to the abstract, we have now followed the journal guidelines (“The abstract should be a single paragraph and should follow the style of structured abstracts, but without headings”) and, thus, erased the heading. Furthermore, slight changes to the abstract were performed in order to clear the study main objectives and potential.
  • Concerning reference/citation styles, we have numbered the references in order of appearance in the text and listed individually at the end of the manuscript, following the journal guidelines.
  • With regard to the tables formatting, we have made and effort to reformat them by assuring that we were following the journal guidelines. Additionally, the information on table 1 may be redundant, as in text we provide the prevalence data on the participants’ filiation. Therefore, we have removed these data from the table, by removing the columns “Researcher” and “Stakeholder”.
  • Finally, concerning the sample representativeness, as this was a feasibility study, the sample representativeness is not considered a matter of concern (differently from efficacy studies, in which samples must be randomized). However, we agree that this study should be replicated using larger samples and, possibly, an RCT study would be of utmost importance to accurately measure the intervention’s efficacy. Still on the sample matter, we come to agree that its nature (participants with significant knowledge on climate change) may be influential on the obtained results. Therefore, we have discussed the abovementioned concerns in the discussion section, as follows:

“Additionally, this study was limited to a sample of European environmental-actors, with relevant knowledge on climate change. Although previous studies have proven that knowledge about climate change does not have a direct effect in promoting behavior change towards more sustainable patterns of living [10], results from this study should be interpreted with cautious taking into account a possible knowledge-related bias effect.   Therefore, future studies should consider more heterogeneous samples, regarding other populations (e.g., school-aged people, vulnerable populations) and other cultures (e.g., non-Western cultures), in order to overcome the likely bias resulting from extensive knowledge about climate change.”

Reviewer 4 Report

The paper “Environment actors confronting a post climate-related disaster scenario: A feasibility study of an action-based intervention aiming to promote climate action” presents a focus group based study to assess how an action-based fieldwork perform in the context of climate change communication.

The study is organized around the post climate-related disaster scenario of a forest fire occurred in central Portugal.

The paper reads well, and it is elaborate. It has a comprehensive introduction about the topic and a good motivation for the work. The structure of the paper is good and the details in the action-based fieldwork and focus group descriptions are adequate.

Below are some comments and clarifications needed:

The authors state that “in order to assess the role of psychosocial variables (e.g., experiential processing) - beyond climate-related knowledge - in enhancing an attitude change regarding climate change, an advanced or expert knowledge the topic was assured in participants’ recruitment”.

From the text it is not clear why the choice of environmental/climate expert as target participants of the study is an essential driving factor to assess psychosocial variables in relation to detect an attitude change regarding climate change. Could this choice introduce bias since this kind of participants have already an extensive knowledge about the climate change. This should be reflected in the text.

The authors use an action based-approach in their study, this method has commonalities with the place-based approach, see papers:

  • Schweizer, Sarah, Shawn Davis, and Jessica Leigh Thompson. "Changing the conversation about climate change: A theoretical framework for place-based climate change engagement." Environmental Communication: A Journal of Nature and Culture1 (2013): 42-62.
  • Carbaugh, Donal, and Tovar Cerulli. "Cultural discourses of dwelling: Investigating environmental communication as a place-based practice." Environmental Communication: A Journal of Nature and Culture1 (2013): 4-23.

The authors could benefit from comparing or acknowledging the similarities between the two processes in their study.

Given the composition of the participants it would be interesting and valuable if the authors highlight in the text the discussion about the relevance of interacting with GIS tools on climate change risk management.

Minor comments:

The  table 1 and 2 needs reformatting since they are not easy to read.

In table 1 the columns “Researcher” and ”Stake-holder” have no values.

Author Response

#Commentary 1: The authors state that “in order to assess the role of psychosocial variables (e.g., experiential processing) - beyond climate-related knowledge - in enhancing an attitude change regarding climate change, an advanced or expert knowledge the topic was assured in participants’ recruitment”. From the text it is not clear why the choice of environmental/climate expert as target participants of the study is an essential driving factor to assess psychosocial variables in relation to detect an attitude change regarding climate change. Could this choice introduce bias since this kind of participants have already an extensive knowledge about the climate change. This should be reflected in the text.

#Authors’ response to commentary 1: Thank you for this commentary with which we totally agree. Despite the fact that cognitive variables tend to not have a direct impact on behavior change processes, we must not dismiss a potential bias effect in our results. Therefore, we have discussed this limitation on the manuscript’s discussion section:

“Additionally, this study was limited to a sample of European environmental-actors, with relevant knowledge on climate change. Although previous studies have proven that knowledge about climate change does not have a direct effect in promoting behavior change towards more sustainable patterns of living [10], results from this study should be interpreted with cautious taking into account a possible knowledge-related bias effect.   Therefore, future studies should consider more heterogeneous samples, regarding other populations (e.g., school-aged people, vulnerable populations) and other cultures (e.g., non-Western cultures), in order to overcome the likely bias resulting from extensive knowledge about climate change.”

#Commentary 2: The authors use an action based-approach in their study, this method has commonalities with the place-based approach, see papers:

Schweizer, Sarah, Shawn Davis, and Jessica Leigh Thompson. "Changing the conversation about climate change: A theoretical framework for place-based climate change engagement." Environmental Communication: A Journal of Nature and Culture1 (2013): 42-62.

Carbaugh, Donal, and Tovar Cerulli. "Cultural discourses of dwelling: Investigating environmental communication as a place-based practice." Environmental Communication: A Journal of Nature and Culture1 (2013): 4-23.

The authors could benefit from comparing or acknowledging the similarities between the two processes in their study.

#Authors’ response to commentary 2: Thank you for suggesting the studies conducted by Schweizer et al. (2013) and Carbaugh et al. (2012). We have carefully read and taken into account the abovementioned studies and discussed our results according to their results. Accordingly, references to the two studies were also added to the references’ list.

#Commentary 3: The  table 1 and 2 needs reformatting since they are not easy to read. In table 1 the columns “Researcher” and ”Stakeholder” have no values.

#Authors’ response to commentary 3: An effort to reformat the tables were conducted, by assuring that the journal guidelines were taken into account. Additionaly, the information on table 1 may be redundant, as in text we provide the prevalence data on the participants’ filiation. Therefore, we have removed these data from the table, by removing the columns “Researcher” and “Stakeholder”.

Round 2

Reviewer 3 Report

The authors have addressed my questions and suggestions from the previous round of review.